# A Metabolomic and Transcriptomic Study Revealed the Mechanisms of Lumefantrine Inhibition of *Toxoplasma gondii*

**DOI:** 10.3390/ijms24054902

**Published:** 2023-03-03

**Authors:** Meiqi Li, Xiaoyu Sang, Xiaohan Zhang, Xiang Li, Ying Feng, Na Yang, Tiantian Jiang

**Affiliations:** 1Key Laboratory of Livestock Infectious Diseases in Northeast China, Ministry of Education, Key Laboratory of Zoonosis, College of Animal Science and Veterinary Medicine, Shenyang Agricultural University, Dongling Road 120, Shenyang 110866, China; 2Department of Pediatrics, School of Medicine, University of California, La Jolla, San Diego, CA 92122, USA

**Keywords:** *Toxoplasma gondii*, lumefantrine, metabolomics, transcriptomics

## Abstract

*Toxoplasma gondii* is an obligate protozoon that can infect all warm-blooded animals including humans. *T. gondii* afflicts one-third of the human population and is a detriment to the health of livestock and wildlife. Thus far, traditional drugs such as pyrimethamine and sulfadiazine used to treat *T. gondii* infection are inadequate as therapeutics due to relapse, long treatment period, and low efficacy in parasite clearance. Novel, efficacious drugs have not been available. Lumefantrine, as an antimalarial, is effective in killing *T. gondii* but has no known mechanism of action. We combined metabolomics with transcriptomics to investigate how lumefantrine inhibits *T. gondii* growth. We identified significant alternations in transcripts and metabolites and their associated functional pathways that are attributed to lumefantrine treatment. RH tachyzoites were used to infect Vero cells for three hours and subsequently treated with 900 ng/mL lumefantrine. Twenty-four hours post-drug treatment, we observed significant changes in transcripts associated with five DNA replication and repair pathways. Metabolomic data acquired through liquid chromatography-tandem mass spectrometry (LC-MS) showed that lumefantrine mainly affected sugar and amino acid metabolism, especially galactose and arginine. To investigate whether lumefantrine damages *T. gondii* DNA, we conducted a terminal transferase assay (TUNEL). TUNEL results showed that lumefantrine significantly induced apoptosis in a dose-dependent manner. Taken together, lumefantrine effectively inhibited *T. gondii* growth by damaging DNA, interfering with DNA replication and repair, and altering energy and amino acid metabolisms.

## 1. Introduction

*Toxoplasma gondii* is an obligate apicomplexan protozoan that has a broad range of hosts. *T. gondii* can infect all warm-blooded animals, including humans, livestock, and wild animals and poses a threat to human and animal health. Thus far, one-third of the human population is chronically infected with *T. gondii* [1]. In China, the human infection rate is about 8% [2,3,4,5,6]. *T. gondii* is an opportunistic pathogen that is lethal to those with compromised immune systems such as AIDS patients, organ transplant recipients, and malignant tumor patients. Infection with *T. gondii* is one of the major causes of death in these patients [1,2,3,4,5,6]. Infection during pregnancy can cause vertical transmission, which could lead to spontaneous abortion, premature birth, or death of the fetus. In newborns, a wide range of birth defects can occur as a result of vertical transmission, including malformation, intracranial calcification, cognitive disorder, hydrocephalus, vision damage, and even death [7,8]. In addition, studies have implicated toxoplasmosis in the mental disorder of 1–10% of psychiatric patients. In particular, toxoplasmosis was linked to schizophrenia [9,10]. In animal husbandry, *T. gondii* infection can cause spontaneous abortion or death of domestic animals including pigs, sheep, and poultry, which causes tremendous economic losses [11]. More importantly, the high incidence of animal toxoplasmosis facilitates *T. gondii* transmission to humans and leads to a high infection rate in humans [12]. Therefore, it is imperative to develop therapeutics to alleviate the suffering inflicted by this notorious pathogen.

To date, treatment of toxoplasmosis relies on pyrimethamine and sulfadiazine, clarithromycin, azithromycin, atovaquone, and epiroprim [13,14,15,16,17,18,19,20]. Pyrimethamine combined with sulfadiazine has been the gold standard therapeutic [17]. Pyrimethamine is an antifolate and acts against dihydrofolate dehydrogenase to affect the synthesis of DNA [16], and sulfadiazine can interfere with folate synthesis. Thereby the combined therapy can act synergistically against *T. gondii* [14]. In rare cases, the therapy can induce side effects such as agranulocytosis, Stevens-Johnson syndrome, and liver necrosis among other adverse effects [18]. The efficacy of clarithromycin and azithromycin in the treatment of *T. gondii* infections in immunocompromised patients has not been confirmed [20]. If these drugs were the last resort to treat toxoplasmosis, they must be used in combination with other drugs such as pyrimethamine. Atovaquone and epiroprim as the second-line drugs act through inhibition of cytochrome bc1 complex and dihydrofolate reductase, respectively [15]. The current treatment period for *T. gondii* is lengthy, ranging from a week to over a year with drugs often showing high toxicity to host cells [19]. Therefore, it is imperative to search for novel drugs that have low side effects and high efficacy and elucidate their mechanisms of action.

One of the drug discovery efforts has been focusing on drug repurposing. Lumefantrine was an antimalarial synthesized by the Beijing Academy of Military Medical Sciences in the 1970s. In the treatment of malaria, lumefantrine has high efficacy, low toxicity, and side effects [21]. Lumefantrine, in combination with artemether, is recommended by WHO to be the first line of antimalarial for different types of malaria, severe or drug-resistant malaria due to its efficacy and safety [22,23]. Our previous study has shown that lumefantrine curtailed the growth of *T. gondii* RH growing in Vero cells in vitro and prolonged the survival time of infected mice [24]. However, despite its significance, the mode of action of lumefantrine is still unknown.

Transcriptomics captures the complete transcript profile to study gene expression and regulation in cells or tissues under a certain biological condition [25,26]. Metabolomics is an innovative tool in drug target identification through accurate quantification of differential metabolites, systematic study of metabolites, metabolic pathways, and cellular metabolism of the parasites [27,28]. Therefore, transcriptome and metabolome can not only explain the “cause” and “effect” of biological processes but also reveal potential drug targets. Tewari et al. used transcriptomic and metabolomic approaches and showed that drug-treated parasites tuned carbohydrate metabolism and reduced metabolite flux through the pentose phosphate pathway, resulting in slower RNA synthesis and increased oxidative stress [29]. Jia et al. analyzed the mechanism of action of anlotinib on colon cancer cell line (HCT-116) through transcriptomic and metabolomic studies. Their results showed that anlotinib affected the protein synthesis of colon cancer cells by regulating amino acid and energy metabolism [30]. These studies demonstrate that a combined analysis of the transcriptome and metabolome can reveal the potential mechanism of the action of a drug.

In this study, we analyzed the overall transcripts and metabolites of lumefantrine-treated *T. gondii* RH strain grown in Vero cells via RNA-sequencing and non-targeted metabolic sequencing technology (LC-MS). We delved into the potential mode of action of lumefantrine in *T. gondii* inhibition. The results of this study shed light on future research to elucidate the drug targets of antiparasitic compounds of various classes.

## 2. Results

### 2.1. Lumefantrine Restricts the Proliferation of T. gondii

To investigate the inhibitory effect of lumefantrine on *T. gondii* growth and its drug target, we designed and carried out four major experiments. We first determined the cytotoxicity of lumefantrine to Vero cells using the Cell Counting Kit-8 (CCK-8). A dose-response assay was carried out and intracellular parasite proliferation was measured via qPCR to determine the appropriate treatment concentration for downstream experiments. Subsequently, we prepared lumefantrine-treated parasites and conducted transcriptome and metabolome analysis. Lastly, we confirmed the apoptosis induced by lumefantrine using the TUNEL assay.

CCK-8 assay was used to test the cytotoxicity of lumefantrine to Vero cells. Lumefantrine was 2-fold serially diluted from 3600 ng/mL to 225 ng/mL. 24 h or 36 h post-infection, lumefantrine did not show significant cytotoxicity to Vero cells (Figure 1A,B). To confirm the inhibitory effect of lumefantrine on *T. gondii* growth, we carried out qPCR and IFA. The qPCR results showed that lumefantrine inhibited 60% of *T. gondii* growth at 900 ng/mL (Figure 1C). IFA results were observed under the confocal fluorescent microscope. The number of parasites per parasite vacuole was recorded by examining at least 100 vacuoles. Lumefantrine-treated (900 ng/mL) vacuoles had a significantly smaller number of parasites (Figure 1D), indicating that lumefantrine inhibits T. gondii proliferation. To visually demonstrate the data from Figure 1D, we created Figure 1E. As shown in Figure 1E, under two different magnifications, most PVs contained 4 or 8 parasites in the drug-treated group, while the PVs containing 8 or 16 were prevalent in the no-drug control.

### 2.2. Lumefantrine Alters T. gondii Gene Expression

To further analyze the effect of lumefantrine on *T. gondii* gene expression, a transcriptome analysis was performed. *T. gondii* were treated with 900 ng/mL lumefantrine for 24 h. Three biological replicates were set up for both the control (control 1, 2, and 3) and the drug treatment groups (LF1, LF2, LF3). To investigate the difference between groups and within groups, we conducted principal components analysis (PCA). The transcriptome profiles of the control groups and the drug-treated groups were separated on the PCA graph above and below PC2 = 0 (Figure 2A). This indicates that there is a significant difference between the drug-treated and non-treated groups. PC1 and PC2 showed 85% and 12% variation, respectively (Figure 2A).

We used DESeq to analyze the 7646 detected genes. Compared with the control group, based on the criteria of *p* < 0.05, fold change >1, one hundred and seventy-five differentially expressed genes (DEGs) were marked as upregulation, and 216 DEGs were downregulated in the drug-treated group (Figure 2B). To verify the accuracy of RNA-seq data, we selected ten differentially expressed genes for qPCR analysis (Figure 2C and Table 1). Based on ToxoDB, these ten genes all have adequate expression levels with expression values (log2) of more than seven. Consistent with the results of RNA-seq, six genes were upregulated, and four were downregulated.

Next, we performed gene ontology (GO) enrichment classification of differentially expressed genes (Figure 2D). Three functional aspects were included in GO analysis, namely, biological process, molecular function, and cellular component. Compared with the control group, biological processes that were significantly affected by lumefantrine included DNA replication (GO:0006260), DNA geometric change (GO:0032392), DNA duplex unwinding (GO:0032508), cell division (GO:0051301), cell cycle (GO:0007049), protein phosphorylation (GO:0006468), and phosphorylation modification (GO:0016310). The molecular functions that were altered by lumefantrine included transferase (GO:0016772), sphingosine N-acyltransferase (GO:0050291), protein kinase (GO:0004672), phosphotransferase (GO:0016773), kinase (GO:0016301), exonuclease (GO:0004527), DNA polymerase (GO:0034061), DNA helicase (GO:0003678), DNA—directed DNA polymerase (GO:0003887), and catalytic activity (GO:0140097) (Figure 2D). Three major cellular components of *T. gondii* were influenced by lumefantrine, namely, spindle (GO:0005819), microtubule organizing center (GO:0005815), and chromosomal region (GO:0098687) (Figure 2D).

KEGG analysis revealed forty significantly enriched pathways. The top twenty most significantly enriched pathways were presented in Figure 2E. Among the most enriched pathways, pathways involved in genetic information processing were significantly impacted including DNA replication, base excision repair, nucleotide excision repair, homologous recombination, and mismatch repair. Pathways involved in toxoplasmosis and arachidonic acid metabolism were also significantly enriched. In the DNA replication and base excision repair, over 20% of the genes were DEGs. In summary, lumefantrine altered the expression of genes involved in DNA replication and repair systems of *T. gondii.*

### 2.3. The Upregulation of Seven Genes Involved in DNA Replication

We classified the top thirty most enriched pathways into four categories based on transcriptomic data (Figure 3A). DNA replication is the most significantly altered pathway under lumefantrine treatment (Figure 3A). Next, we zoomed in on the DEGs in the DNA replication pathway. Among the thirty genes annotated in DNA replication, seven genes were found to be upregulated (*p* < 0.05) (Figure 3B). They are proliferating cell nuclear antigen PCNA2 (TGME49_320110), DNA polymerase (TGME49_280690, TGME49_268600, and TGME49_233820), a putative DNA replication licensing factor MCM2 (TGME49_214970), a putative DNA replication licensing factor MCM4 (TGME49_219700), and a putative helicase (TGME49_261850).

### 2.4. Lumefantrine Interfered with T. gondii Metabolism

To investigate how lumefantrine alters *T. gondii* metabolism, we performed liquid chromatography-tandem mass spectrometry (LC-MS) analysis. A total of 432 differential metabolites were identified using positive and negative ion mode LC-MS. To analyze differentially accumulated metabolites between groups, we used orthogonal Partial Least Squares Discriminant Analysis (OPLS-DA) (Figure 4A,B). OPLS-DA score plots showed the separation of the drug-treated group and the non-treated group in both positive and negative ion modes, which indicated that the metabolite profile in the drug-treated group was significantly different from that in the control group. As shown in Figure 4C, the overall distribution of DAM was relatively symmetrical. In particular, the biosynthesis of 29 metabolites was significantly affected, as shown in the volcano and heat maps shown in Figure 4C,E. Among the downregulated metabolites was thymidine. Thymidine exists exclusively in DNA and the T-loop of tRNA. Among the upregulated metabolites were cytidine and D-fructose-1,6-diphosphate. The former is a component of RNA and a precursor of uridine which is used in RNA synthesis and the latter is involved in glycolysis which affects the energy metabolism of *T. gondii*. Differentially accumulated metabolites were annotated to KEGG pathways. The KEGG analysis showed that differential metabolites were mainly enriched in 27 pathways (Figure 4D). The significantly impacted pathways include galactose metabolism, fructose and mannose metabolism, sulfur relay system, arginine, and proline metabolism, ABC transporters and phenylalanine, tyrosine, and tryptophan biosynthesis. Most of these pathways are related to energy and amino acid metabolisms. In summary, our data suggest that lumefantrine induced the metabolic disorder of *T. gondii*, affecting glucose catabolism and amino acid metabolism.

### 2.5. Lumefantrine-Induced T. gondii Apoptosis

To elucidate whether lumefantrine induces apoptosis, we used the TUNEL method to detect the integrity of DNA strands of lumefantrine-treated *T. gondii*. The red fluorescent signal increased with the lumefantrine concentration (900 ng/mL vs. 1800 ng/mL), while no fluorescent signal was observed in the control group (Figure 5A). This indicates that lumefantrine-induced DNA breakage of *T. gondii*. Quantitative analysis using confocal microscopy showed that the apoptosis rates in the drug-treated groups were significantly higher than that in the control. In addition, lumefantrine-induced apoptosis was dose-dependent as the apoptosis rate was significantly higher in *T. gondii* treated with 1800 ng/mL of lumefantrine compared to that with 900 ng/mL of lumefantrine (Figure 5B).

## 3. Discussion

The current work investigated the mechanism of the action of lumefantrine on *T. gondii*. Our data showed that lumefantrine-induced apoptosis in *T. gondii* interfered with DNA replication and repair, and caused metabolic alterations. We first studied the cytotoxicity of lumefantrine to Vero cells and determined the appropriate drug concentration for *T. gondii* treatment. We found that lumefantrine at 900 ng/mL reduced 60% of the intracellular proliferation of *T. gondii*. Transcriptomic data showed that lumefantrine altered transcripts involved in DNA replication and repair and caused metabolic changes.

We found 41 differential pathways through transcriptomic study including 12 pathways that were DNA replication related, 5 RNA-associated pathways, and 23 pathways that were related to energy metabolism. The top three significantly impaired pathways were DNA replication, base excision repair, and nucleotide excision repair. *T. gondii* can propagate in a wide array of cell types and replicate every 6–8 h. Precise duplication of DNA ensures the sustainability and stability of *T. gondii* genetic material [31]. Base excision repair is the major mechanism of DNA repair, through which mutated bases or nucleotides were removed [32]. The highly conserved nucleotide excision repair system is used to restore genome integrity including repairing hydrogen bonds between strands [33]. In addition, it contributes to promoting mRNA synthesis or shaping the 3D architecture of chromatin [34].

Among the significantly altered pathways were DNA replication, base excision repair, and nucleotide excision repair. Proliferating cell nuclear antigen (PCNA) is an auxiliary protein of DNA polymerase δ and ε and is central to DNA replication and repair. Lima et al. have shown through transcriptome data that *T. gondii* infection-induced gene expression changes related to DNA replication and repair [35]. Their data showed that in *T. gondii*-infected human neutrophils, the PCNA transcript was upregulated which was consistent with our finding in lumefantrine-treated *T. gondii*. In their study, the PCNA transcript upregulation was linked to the delayed apoptosis of *T. gondii*-infected neutrophils. Minichromosome maintenance protein 2 (MCM2) and MCM4 play critical roles in DNA replication initiation. Recent studies have shown that MCM proteins play roles in replication elongation and genome stability [36]. MCM OB domain and MCM2/3/5 family were significantly affected by the deletion of *T. gondii* UBL-UBA shuttle protein which was found to regulate DNA replication [37]. We found the upregulation of PCNA, MCM2, MCM4, and DNA polymerase. It has been shown that DNA damage activated PCNA, and PCNA was involved in nucleotide excision repair, base excision repair, and mismatch repair [38]. DNA repair also requires the participation of DNA polymerase δ and/or ε. The upregulation of these proteins in our study implies DNA damage induced by lumefantrine treatment.

The DNA-damaging effect discovered in this study is not uncommon among drugs. For example, artemisinin induced a DNA-damaging effect in *Plasmodium falciparum*, similar to the effect of methyl methanesulphonate, an alkylating agent [39]. In a recent study, artemisinin resistance was found to be associated with enhanced DNA damage repair [40]. The molecular mechanism of the DNA damage induced by lumefantrine in *T. gondii* is, however, yet to be investigated. It has been shown that the DNA damage induced by artesunate in *P. falciparum* was accompanied by an increased level of ROS (Reactive Oxygen Species) in the parasites and that artesunate exerted DNA breakage in a dose- and time-dependent manner [41]. Albendazole was shown to arrest the cell cycle at the G2/M phase in *Giardia duodenalis* and induced nuclei acid oxidative damage evidenced by the phosphorylation of histidine H2AX [42].

Among the pathways involving significantly enriched DAMs, the most significantly impacted is galactose metabolism. Galactose can be converted to glucose, lactose, and other sugar intermediates which can participate in a series of energy-related metabolic processes. It enters glycolysis through conversion to glucose-1-phosphate (G1P) [43]. Secondly, the ABC transporter pathway was significantly altered. The ATP-binding cassette (ABC) transporter proteins are a superfamily of membrane proteins that are responsible for the ATP-powered transportation of a wide range of substances [44], including cellular metabolites, drugs, lipids, and sterols. Multidrug-resistant protein 1 (MDR1), a member of the ABC transporter superfamily, was found to contribute to drug resistance in malaria. In the case of artemisinin and mefloquine, MDR1 delivers drugs into the digest vacuoles, preventing them from hitting the drug target. In the case of chloroquine, MDR1 transports the compound out of its site of action—the digestive vacuole. Mutations and copy number variations in MDR1 have been associated with drug resistance in Plasmodium falciparum [45]. The downregulation of ABC transporter is frequently found in drug resistance in cancer cells [30,46]. Out of the 13 ABC membrane transporters in *T. gondii*, TgABC.B1 is the most expressed and shows the highest similarity to the human MDR1 protein. The participation of this transporter in drug resistance in *T. gondii* has been proposed [47]. Our data suggest that lumefantrine altered the transportation of various metabolites through the ABC transporter. ABC transporter pathway could possibly participate in lumefantrine transportation in *T. gondii*. Thirdly, the fructose and mannose metabolism pathway was significantly affected which led to the poor utilization of host-sourced glucose and ultimately slowed down the replication of *T. gondii* [48]. The fourth noteworthy pathway is arginine and proline metabolism. Since *T. gondii* is an arginine auxotroph, the reduced arginine production in the parasite could restrict parasite reproduction [49]. Modulation of arginine metabolism toward ornithine, proline, and polyamines can be viewed as a parasite adaptation. The diversion of arginine metabolism toward arginase degradation and the reduction of iNOS- induced nitric oxide have been shown to curtail parasite proliferation [50]. In summary, the metabolic data showed that lumefantrine disturbed the sugar and amino acid metabolisms of *T. gondii,* especially galactose and arginine.

## 4. Materials and Methods

### 4.1. Cells and Parasites

Vero cells were cultured at 37 °C, 5% CO_2_ incubator in DMEM medium (MACGENE, Beijing, China) containing 100 U/mL penicillin, 100 μg/mL streptomycin (MACGENE, China), and 8% heat-inactivated fetal bovine serum (BI, Uruguay, Israel). *T. gondii* RH strain was cultured in DMEM medium supplemented with penicillin, streptomycin, and 2% heat-inactivated fetal bovine serum (BI, Uruguay, Israel).

### 4.2. Cytotoxicity Assay

The cytotoxicity of lumefantrine (Sigma, Ronkonkoma, NY, USA) to Vero cells was evaluated using the Cell Counting Kit-8 (CCK-8) (Baisai, China). Vero cells (3 × 10^4^) were seeded in each well of a 96-well plate and cultured in DMEM for 24 h. Lumefantrine was 2-fold serially diluted in DMEM from 3600 ng/mL to 225 ng/mL. The blank control was 110 μL of DMEM, and negative control was Vero cells with 110 μL of DMEM. After culturing for 24 or 36 h, 10 μL of CCK solution was added into each well and incubated for 1 h. Optical density (OD) was measured at 450 nm using a microplate reader.

### 4.3. Intracellular Proliferation Assay

Immunofluorescent assay (IFA) was used to quantify the proliferation. *T. gondii* was inoculated onto Vero cells growing on slides inserted in a 12-well plate and allowed to infect for 2 h at 37 °C, 5% CO_2_. Cells were washed, supplied with serum-free medium or medium containing 900 ng/mL lumefantrine, and incubated for 20 h. Cells were then fixed with 4% paraformaldehyde for 15 min, permeabilized with 0.25% Triton X-100 for 15 min, and blocked with 3% bovine serum albumin for 30 min. Rabbit anti-TgALD was added as primary antibody and incubated for one hour. The secondary antibody (Alexa Fluor488 goat anti-rabbit) was added subsequently and incubated for one hour. Nuclear DNA was stained with DAPI for 10 min. Image and data acquisition was performed using Leica fluorescence microscope system (Leica, Wetzlar, Germany) at 189× and 124× magnifications. At least 100 parasite vacuoles per sample were examined to determine the number of parasites per vacuole.

### 4.4. In Vitro Proliferation Assay

The inhibitory effect of lumefantrine on *T. gondii* was detected by qPCR. Vero cells were inoculated into a 6-well plate and allowed to grow for 12 h. 3 × 10^5^ of newly released RH tachyzoites were used to infect Vero cells for 4 h at 37 °C. Cells were washed twice to rid of extracellular parasites. Parasites and cells were treated with lumefantrine (0, 900, and 1800 ng/mL) in DMEM for 24 h. Cells were scraped off and used for DNA extraction using a DNA extraction kit (Tiangen, Beijing, China). The proliferation of parasites was detected using the 2^−∆∆t^ relative expression method. The primers targeting *T. gondii* GAPDH (housekeeping gene) were GAPDH-F (ATTTTGCTTGGGATTCGAGGA) and GAPDH-R (TGCAGGGTAACGATCAAAAAATG).

### 4.5. Transcriptome Sample Preparation

Three sets of transcriptome samples were prepared. Vero cells grown in T25 flasks were infected with 1 × 10^7^ tachyzoites for 3 h at 37 °C, 5% CO_2_. Media was replaced with serum-free DMEM with or without drug (900 ng/mL lumefantrine). After incubation for 24 h, cells were scraped off and syringed with a 22 G ½ inch needle on a 5 mL syringe to liberate parasites. We then passed the mixture through 3 μM of filter to obtain the parasites. After flash freezing in liquid nitrogen, cell lysates were ready for library preparation.

The mRNA with polyA structure in the total RNA was enriched by Oligo(dT) magnetic beads. RNA was fragmented into pieces of roughly 300 bp. Using RNA as a template, the first strand of cDNA was synthesized with 6-base random primers and reverse transcriptase, and the second strand of cDNA was synthesized using the first strand of cDNA as a template. After the library was constructed, PCR amplification was used to enrich the library fragments. Fragments of around 450 bp were selected. Agilent 2100 Bioanalyzer was used for quality inspection, through which the total and effective concentrations of the library were obtained. The libraries containing different indexes were pooled which were then diluted to 2 nM and denatured by alkali. The pooled libraries were subjected to pair-ended Illumina sequencing.

### 4.6. Transcriptome Data Validation

Among all differentially expressed genes, 10 genes identified by RNA-seq analysis were selected for validation by qRT-PCR. Samples preparation procedure was the same as described in Section 4.5. Total RNA was extracted using TRIzol, and then cDNA was synthesized from total RNA using PrimeScriptTM II First Strand cDNA Synthesis Kit (Takara, Dalian, China) according to the manufacturer’s instructions.

All qRT-PCR experiments were performed in three technical replicates using GAPDH as the reference gene. The qRT-PCR primers used in this study were described in Table 1. The cycle conditions were 95 °C for 5 min, 40 cycles of 95 °C for 10 s, 60 °C for 10 s, and 72 °C for 15 s, and the melting curve temperature is 72~95 °C. Gene expression was calculated using the 2^−∆∆t^ relative expression method.

### 4.7. Metabolome Sample Preparation and Metabolite Extraction

Seven groups of metabolome samples were prepared. Parasites were prepared in the same manner as shown in Section 4.5. After flash freezing in liquid nitrogen, parasites were mixed with tissue extraction solution (75% of methanol and chloroform at 9 to 1 ratio, 25% H_2_O), three steel balls, and ground in a high-throughput tissue grinder at 50 Hz for 60 s. The grinding process was repeated twice. The cell culture was then subjected to ultrasonication at room temperature for 30 min, subsequently placed on ice for 30 min, and centrifuged at 12,000 rpm, 4 °C for 10 min. Notably, 850 μL of the supernatant was taken and dried in a vacuum concentrator. In addition, 200 μL of 2-chlorobenzalanine solution made with 50% acetonitrile solution was added to redissolve the samples which were subsequently filtered through a 0.22 μm membrane. Notably, 20 μL of the filtrate from each sample was pooled into a QC sample to be used for data normalization. The remaining samples were used for LC-MS detection.

### 4.8. LC-MS/MS Analysis

LC-MS/MS was carried out with positive and negative ion mode electrospray ionization. Chromatographic separation was performed using ACQUITY UPLC^®^ HSS T3 1.8 μm (2.1 × 150 mm) chromatographic column with the temperature of the autosampler set at 8 °C, the flow rate at 0.25 mL/min, and the column temperature at 40 °C. 2 μL of the sample was injected for gradient elution. The positive ion mobile phase consisted of 0.1% formic acid (solvent C) and 0.1% formic acid in acetonitrile (solvent D), and the negative ion mobile phase was comprised of 5 mM of ammonium formate (solvent A) and 20% acetonitrile (solvent B). The gradient elution program was as follows: 2% B/D (0~1 min), 2~50% B/D (1~9 min), 50~98% B/D (9~12 min), 98% B/D (12~13.5 min), 13.5~14 min, 98~2% B/D(13.5~14 min), 2% D—positive mode (14~20 min) or 2% B—negative mode (14~17 min). Ionization was accomplished with electrospray ionization source (ESI) in positive and negative ionization modes. Mass spectrometry was conducted with a positive ion spray voltage of 3.50 kV, a negative ion spray voltage of 2.50 kV, sheath gas of 30 arbs, and auxiliary gas of 10 arbs. The capillary temperature was 325 °C, full scan was performed with a resolution of 70,000, and the scan range was 81–1000. HCD was used for secondary decomposition, and the collision voltage was 30 eV. Dynamic exclusion was used to remove unnecessary MS/MS information.

The obtained raw data were converted into mzXML format by Proteowizard software (v3.0.8789). The XCMS package of R (v3.3.2) was used for peak identification, peak filtration, and peak alignment including the following parameters, bw = 5, ppm = 15, peak width = c (5, 30), mzwid = 0.015, mzdiff = 0.01, method = “centWave”. A data matrix was obtained including mass-to-charge ratio (*m*/*z*), retention time, and peak area (intensity). 20,085 precursor molecules were obtained in positive ion mode, and 10,983 precursor molecules were obtained in negative ion mode, and the data were exported to excel for subsequent analysis. To make the data comparable, batch normalization of the peak area was performed. Orthogonal Partial Least Squares Discriminant Analysis (OPLS-DA) was used for multivariate statistical analysis using software package, SIMCA-P (v13.0) and the ropls package R. The data set was scaled using the pheatmap package of R (v3.3.2) by which a hierarchical clustering map of relative quantitative values of metabolites was obtained.

### 4.9. Lumefantrine-Induced DNA Damage

TUNEL test can be used to detect DNA breakage in the final stage of apoptosis [51]. *T. gondii* was cultured in Vero cells for 48 h with 0, 900, or 3600 ng/mL of lumefantrine. Parasites were released from host cells with a 22 G ½ inch needle in a 5 mL syringe. After filtration through 3 μM filter, parasites were centrifugated at 1200 g for 5 min and resuspended with 200 μL of DMEM medium. Slides were inserted into a 24-well plate and treated with 200 μL of poly-L-lysine at 37 °C for 30 min. After repeated washing, parasites were added to the slides and incubated for 20 min. Parasites were fixed, premetallized, blocked, and stained with primary and secondary antibodies in the same fashion as described in Section 4.3. Hoechst solution (diluted 1:1000) was used for nuclei staining. Notably, 50 μL TUNEL detection solution (5 μL of TdT enzyme, 45 μL of fluorescent labeling solution, Biyuntian, China) was added to the slides and incubated at 37 °C in an immunohistochemical wet box for 60 min. After washing, the slides were sealed and observed under a fluorescent microscope. The nuclei of TUNEL-positive parasites should appear red.

The ratio of the number of DNA-damaged parasites (red fluorescence) to the total number of tachyzoites (green fluorescence) was defined as the apoptotic rate. Ten fields were randomly selected for data acquisition and three technical replicates were included.

### 4.10. Statistical Analysis

Statistical analysis was performed using GraphPad Prism 8.0.2 software. All values were expressed as mean ± S.D. All data were analyzed using ANOVA or *t*-test. *p* values less than 0.05 were considered statistically significant.

## 5. Conclusions

In summary, we identified the potential drug targets of lumefantrine on *T. gondii* through transcriptomic and metabolomic studies. Our data suggest that lumefantrine likely exerts its inhibitory effect on *T. gondii* through damaging DNA, impairing DNA replication and repair, and inducing metabolic alterations to hinder the efficient acquisition of energy and essential amino acids.

## Figures and Tables

**Figure 1 ijms-24-04902-f001:**
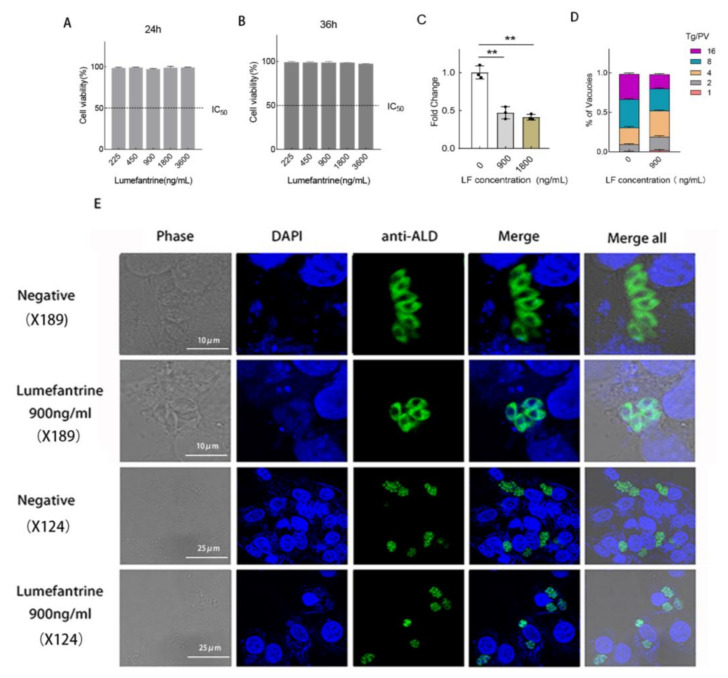
The cytotoxicity of lumefantrine and its inhibitory effect on *T. gondii*. (**A**) Cytotoxicity of lumefantrine to Vero cells after 24 h. Compared to DMEM group, no significant cell viability change was detected. Three technical replicates and two biological replicates were set up and data were represented as average ±S.D. (**B**) Cytotoxicity of lumefantrine to Vero cells after 36 h. No significant difference was observed as compared to the DMEM control. Three technical replicates and two biological replicates were set up and data were represented as average ±S.D. (**C**) Dose-response assay via qPCR. *T. gondii* was allowed to infect Vero cells for 4 h and subsequently treated with different concentrations of lumefantrine for 24 h. qPCR was conducted to detect T. gondii proliferation. Three technical replicates and two biological replicates were set up and data were represented as average ±S.D. (** *p* ≤ 0.01). (**D**) Enumeration of *T. gondii* parasitophorous vacuoles under the microscope following lumefantrine treatment. Three technical replicates and two biological replicates were set up. Data were represented as average ±S.D. (**E**) Confocal microscopy examination of lumefantrine-treated *T. gondii*. The top two rows of images were magnified 189 times and the bottom two were zoomed-out images of the top two rows with a magnification of 124 times. Anti-TgALD antibody was used to stain the cytosol of *T. gondii*. DAPI was used to stain nuclei. The negative controls were groups without drug treatment.

**Figure 2 ijms-24-04902-f002:**
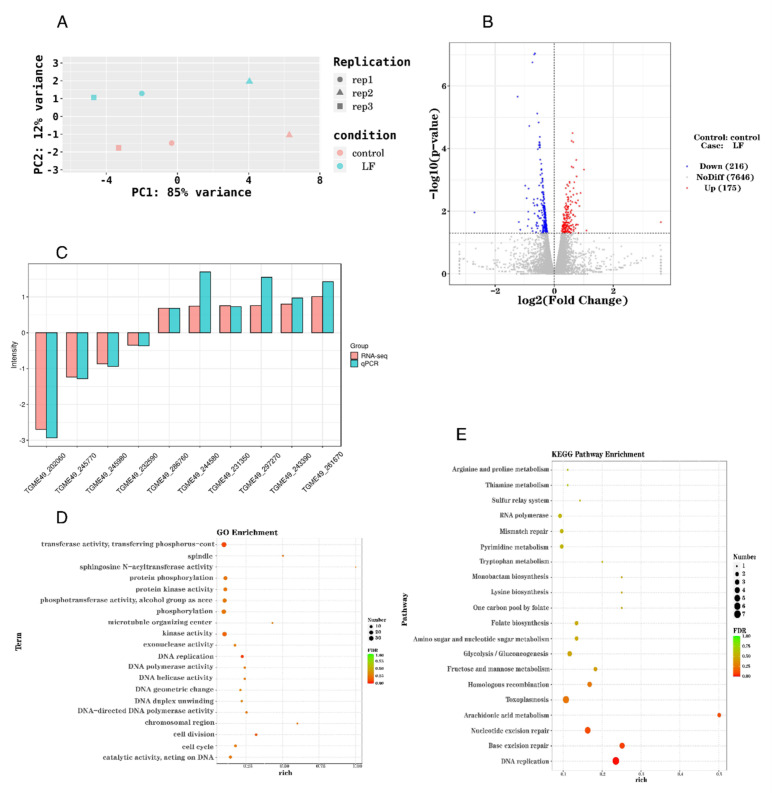
Transcriptome analysis of the effect of lumefantrine on *T. gondii*. (**A**) Transcriptome profile differences between groups and within groups were analyzed via principal components analysis (PCA). LF1, LF2, and LF3 are the three biological replicates of the drug treatment group. Control 1, 2, and 3 are three biological replicates of the control group without drug treatment. PC1, principal component 1; PC2, principal component 2. (**B**) The volcano map of differentially expressed genes. In this map, genes that were upregulated, downregulated, or not significantly affected were mapped. *X*-axis is the log2 fold change and the *y*-axis is the −log10(*p*-value). The horizontal line is the threshold of *p*-value = 0.05. Red dots indicate gene upregulation and blue dots indicate gene downregulation. Gray dots denote genes with no significant change. (**C**) Comparison between transcriptome data and qPCR data of the selected ten differentially expressed genes. Based on transcriptomic data, ten GEOs with high expression levels were selected for qPCR analysis. (**D**) Bubble chart of gene ontology (GO) enrichment analysis of differentially expressed genes. FDR (false discovery rate) ranges from zero to one. The closer FDR is to zero, the stronger the enrichment. The size of the dot denotes the number of DEGs. Rich factor is the ratio of the number of the DEGs to the total number of annotated genes. (**E**) Bubble chart of KEGG pathway analysis. The *y*-axis is the different KEGG pathways. The *x*-axis is the rich factor. Rich factor is the ratio of the number of the DEGs to the total number of annotated genes in this pathway. The size of the dot correlates with the number of DEGs annotated in the pathway.

**Figure 3 ijms-24-04902-f003:**
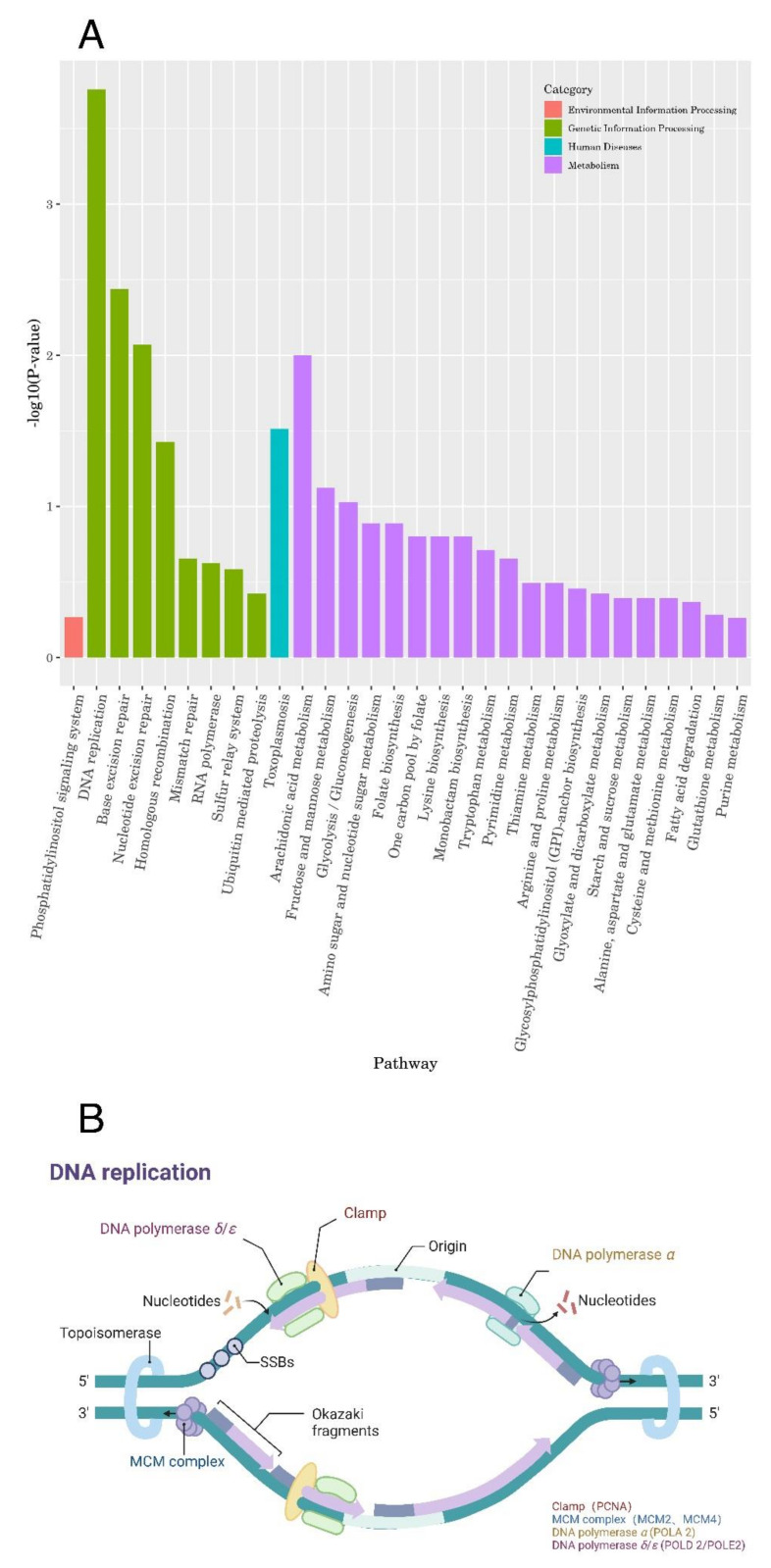
Transcriptome data suggests that lumefantrine interfered with *T. gondii* DNA replication. (**A**) The classification of the thirty enriched pathways. The top thirty most enriched pathways were grouped into four categories, namely, environmental information processing, genetic information processing, human diseases, and metabolism. (**B**) The seven genes that were upregulated in the DNA replication pathway. We illustrated the component of DNA replication using BioRender (created with BioRender com). The red components were proteins that were encoded by the upregulated genes identified in this study.

**Figure 4 ijms-24-04902-f004:**
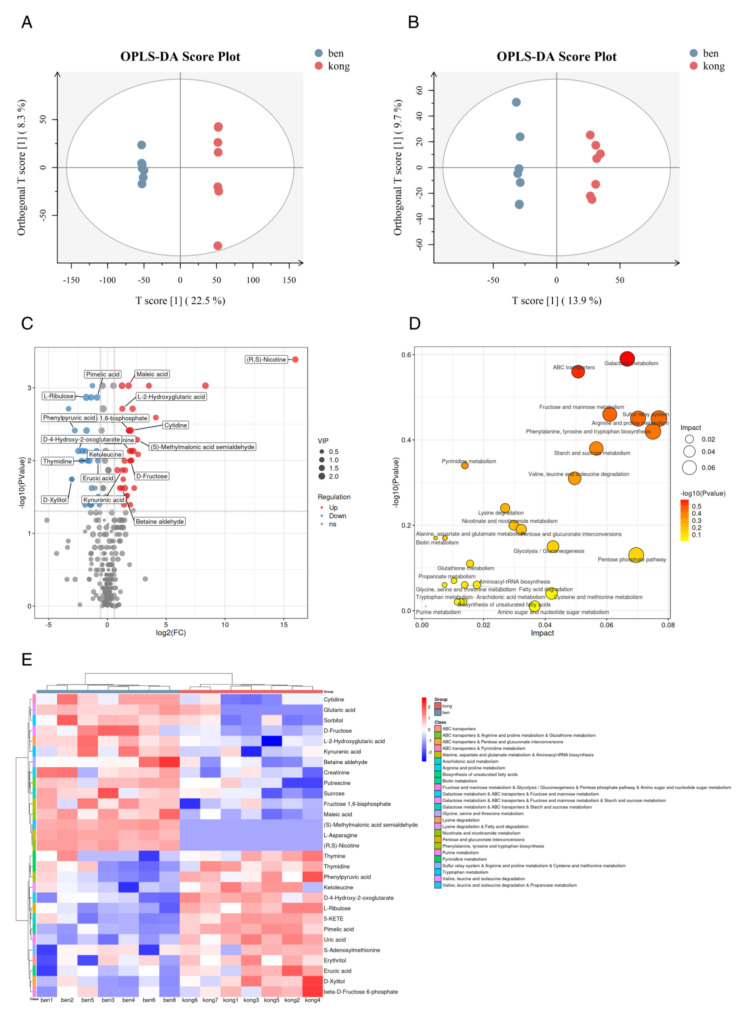
The metabolomic analysis of lumefantrine activity in *T. gondii*. (**A**) OPLS-DA of metabolite samples using electrospray ionization positive ion mode. OPLS-DA: orthogonal Partial Least Squares Discriminant Analysis. Seven replicates were included for both the drug-treated group and the non-treated group. The drug-treated group was denoted as “ben” and the non-treated group was denoted as “kong”. (**B**) OPLS-DA of metabolite samples using electrospray ionization negative ion mode. Seven replicates were included for both the drug-treated group and the non-treated group. (**C**) Volcano map of differentially accumulated metabolites (DAMs). The log2 of fold change was plotted against the −log10 of *p*-value. The two vertical dotted lines in the figure were the 2-fold change threshold, and the horizontal dotted line was the *p*-value = 0.05 threshold. Red and blue dots represented up- and downregulation of DAMs, respectively, and gray dots represented non-significant metabolites. The VIP score was listed to the right of the graph ranging from 0.5 to 2. A variable with VIP score close to or greater than one was considered important in the model. VIP: Variable importance in projection. (**D**) Bubble chart of KEGG pathways involving significantly enriched DAMs. Twenty-seven pathways were shown. The rich factor is the proportion of the number of DAMs to the total number of enriched genes. The size of the circle loosely represented the number of DAMs. The yellow to red color scheme represented −log10 of *p*-value. The impact score listed to the right indicated the impact of the DAMs on the pathway with higher score indicating higher impact. (**E**) The cluster heatmap of DAMs to show the metabolic differences between lumefantrine-treated and non-treated *T. gondii*. Metabolites with similar metabolic patterns were clustered to infer the biological functions of known or unknown metabolites. The red and blue colors denoted up- and downregulation of DAMs, respectively. Impact factor was the ratio of the number of DAMs to the total number of metabolites in the pathway.

**Figure 5 ijms-24-04902-f005:**
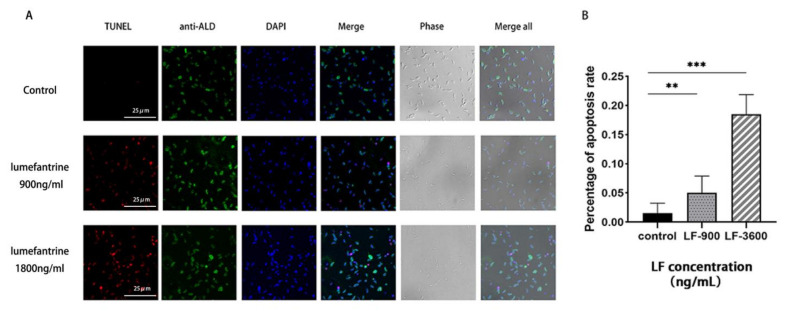
TUNEL assay to investigate apoptosis induced by lumefantrine. (**A**) Observation of DNA breakage under confocal microscopy caused by lumefantrine treatment. RH *T. gondii* in Vero cells were treated with lumefantrine (900 ng/mL or 1800 ng/mL) for 24 h and harvested for TUNEL test. Red signal indicated DNA damage. Green fluorescence was the result of staining of *T. gondii* cytoplasm using anti-TgALD antibody. The blue fluorescence was nuclei staining with DAPI. (**B**) Determination of the apoptosis rate using TUNEL. The ratio of the number of DNA-damaged parasites to the total number of parasites was defined as the apoptotic rate. With the increase of lumefantrine concentration, the apoptosis rates increased. Data shown here were the averages of the three biological repeats. The error bar was the standard deviation (** *p* ≤ 0.01, *** *p* ≤ 0.001) and *t*-test was used for statistical analysis.

**Table 1 ijms-24-04902-t001:** qPCR primers used to verify the selected ten differentially expressed genes identified in RNA-seq analysis.

Gene ID	Gene Name	Forward Primer	Reverse Primer
TGME49_202060	hypothetical protein	CCTCACCAAGGAATGACACGA	TATGACGCTGGAGCGAGGAA
TGME49_245770	hypothetical protein	GAACATGCGAATGCACGAGA	ATGCGTTCCCCCACTTTTTG
TGME49_245980	hypothetical protein	CCACGTGTAATGCGTGAGTTG	TCACACTGGTACGACAAGCAA
TGME49_232590	glutamate-cysteine ligase, catalytic subunit domain-containing protein	CCCCTGGTGACAGCTGATTT	TTGGTCGTGATAGGGCGATG
TGME49_286760	hypothetical protein	AACGGGTTGAGTGTGCAAGA	GTCGTTTGAGGCTGCTGTTG
TGME49_244580	L1P family of ribosomal protein	AGCAGCCAGAGTGTTTCAGT	CAGAGCGCGGGAAGAGTAAA
TGME49_231350	glucosamine-fructose-6-phosphate aminotransferase	GATCGCACCCTACCAGAACC	TGAGCAGTTCGCCGTAGATG
TGME49_297270	hypothetical protein	TGGGGATACCCCATTTTCGG	TGTCTCATACTCCCAGGCCA
TGME49_243390	hypothetical protein	ATCGAGGCATCATCAACCCC	CGGGTAGGCAAGGGATCAAA
TGME49_261670	ribonuclease H1/H2 small subunit protein	CCACTTTCTCACCATGCCCT	GAACCATCCGTTTCCCCTGT

## Data Availability

The data presented in this study are available upon request from the corresponding author. The RNA-Seq dataset described in this study has been deposited in the NCBI Short Read Archive Database (https://www.ncbi.nlm.nih.gov/sra/PRJNA917029, accessed on 2 January 2023). Registration number is PRJNA917029.

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
