# Peer review of "A Metabolomic and Transcriptomic Study Revealed the Mechanisms of Lumefantrine Inhibition of Toxoplasma gondii"

_ijms, 2023, doi:10.3390/ijms24054902_

Round 1
Reviewer 1 Report
Overall the manuscript is good, well organized and even I trust it is possible to improve the manuscript than its present form. Hence I advise the authors to take some of their time to revise& improve the paper . For more information please see the attached herewith file.

Author Response
General comments
Authors should try all their best to significantly improve the English language of the
Manuscript.
Misspellings, many types of mistakes and all other sorts of errors in the paper should
be carefully rechecked and recorrected.
Response: Thank you for your good suggestions. We have revised the manuscript to improve the English language.
Abstract
Line 21: please write in full the abbreviation “RH”
Response: Thank you for your good suggestions. RH are the initials of a baby boy from whom the strain (RH88) was isolated. The RH strain has been used across the globe by T. gondii researchers for almost fourty years. It’s well adapted to grow in cell culture in vitro and replicates fast with a 2-day life cycle. Its virulence and genetic information have been well studied that every T. gondii scholar is familiar with it. It’s known as “RH88” or simply “RH” , for example, the reference “Tosetti, N., Dos Santos Pacheco, N., Soldati-Favre, D. & Jacot, D. Tree F-actin assembly centers regulate organelle inheritance, cell-cell communication and motility in Toxoplasma gondii. eLife 8, e42669 (2019)”, and so on.
Introduction
This section is good
Response: Thank you very much!
Results
Line 100-109: Authors need either deletion or rewrite of from line 100-109 as all
the sentences repat what is already available in the materials & methods section.
This types of problems need to be corrected on other pages of this section.
Response: Thank you for your good suggestions. We have revised them as “4.1., 4.2., 4.3........4.10.” in “4. Materials and Methods” in the revised manuscript.
Line 114: Authors should rewrite the sentence as “The number of parasites per
parasite vacuole were recorded by examining at least 100). ”
Some of the figures are not clear enogh for the readers of this paper.
Response: Thank you for your good suggestions. We have revised the sentence as “The number of parasites per parasite vacuole were recorded by examining at least 100 vacuoles” on line 115-116 in the revised manuscript.
Materials and Methods
This section is good improvement of English
Study area and materials------ need to be clearly indicated for the readrs of this paper.
Duration or time of the study was not provided, too.
Response: Thank you for your suggestion. We have added some information of the study in the materials and methods session 4.4. and 4.5. on line 411, 417-419.
Discussion
Need significant revisions
Response: Thank you for your good suggestions. We have revised the discussion on line 333-343 in the revised manuscript.
References
Authors need to avoid old references.
Response: Thank you for your good suggestions. We have deleted reference 32 and added more references as we work to strengthen the discussion.
Add references:
- Gupta, D.K., Patra, A.T., Zhu, L., Gupta, A.P., Bozdech, Z. DNA damage regulation and its role in drug-related phenotypes in the malaria parasites. Sci Rep2016, 6, 23603.
- Xiong A., Prakash, P., Gao, X.H., Chew, M., Tay, I.J.J., Woodrow, C.J., Engelward, B.P., Han, J.Y., Preiser, P.R. K13-Mediated Reduced Susceptibility to Artemisinin in Plasmodium falciparumIs Overlaid on a Trait of Enhanced DNA Damage Repair. Cell Rep 2020,5, 107996.
- Gopalakrishnan A.M., Kumar, N. Antimalarial Action of Artesunate Involves DNA Damage Mediated by Reactive Oxygen Species. Antimicrob Agents Ch2015, 1, 317-325.
- Rodrigo, M.E., Raúl, A.G., Emma, S., Guadalupe, O.P.Albendazole induces oxidative stress and DNA damage in the parasitic protozoan Giardia duodenalis. Front Microbiol 2015, 6, 800.)

Reviewer 2 Report
The topic is very important. Introduction nicely introduce the topic to the audience. Method is adequate. Results are well described. Discussion has sufficient argument. Conclusions have been drawn on the basis of solid data.
Author Response
Thank you for your recognition of our work !
Reviewer 3 Report
Comments to the Authors
In this work, Li et. al. used metabolomics and transcriptomics to investigate the mechanisms of lumefantrine inhibition against T. gondii. They found that the drug inhibited the growth of the pathogen by damaging DAN repair and replication and altering the metabolisms of energy and amino acids. This study provided some important insights into the mode of action of lumefantrine. However, there are some issues that authors may need to address.
1. The authors included drug treatment and control group to analyze their results. However, in an attempt to identify the potential drug target of lumefantrine, it would be better to also include a positive control using pyrimethamine whose target is relatively well-known. A PCA plot then can provide some useful information how different mechanism the two drugs work. If not, the authors might need to dig into the published transcript data of pyrimethamine on the protozoan. By comparison, the molecular mechanism of lumefantrine would be cleared.
2. For RNA-seq, please specify how T. gondii was grown and harvest (Line 138, Line 401). How was the host RNA from Vero cell removed?
3. Please add scale bars to Figure 1E, even though the microscopic magnification was mentioned.
4. How the qPCR data were normalized? An internal housekeeping gene was required in each samples to normalize the levels of your genes of interest.
5. The PCA plot in Figure 2A only had three samples for each group. To increase the power, it requires at least 4 samples in each group. Someone might argue that the left four points are grouped tighter more closer than the two sample on the right. It is also more informative if the authors included at least one more treatment over the time scale. This would help us know which genes are first impacted by the drug treatment.
6. Please make the font size bigger for Fig. 2CDE, Fig. 3A, Fig. 4 CDE. Especially the heatmap scale bar in Fig 4E.
7. It seems that lumefantrine affected the DNA replication system of the protozoan but why did it not affect Vero cells (Fig 1)? Can the authors discuss a little bit more? Maybe the drug cannot get into the cell nucleus?
8. Could the author also explained why the galactose metabolism was affected (increased or decreased) by lumefantrine? The DMEM medium seems not to contain galactose.
Author Response
Reviewer 3:
In this work, Li et. al. used metabolomics and transcriptomics to investigate the mechanisms of lumefantrine inhibition against T. gondii. They found that the drug inhibited the growth of the pathogen by damaging DAN repair and replication and altering the metabolisms of energy and amino acids. This study provided some important insights into the mode of action of lumefantrine. However, there are some issues that authors may need to address.
1.The authors included drug treatment and control group to analyze their results. However, in an attempt to identify the potential drug target of lumefantrine, it would be better to also include a positive control using pyrimethamine whose target is relatively well-known. A PCA plot then can provide some useful information how different mechanism the two drugs work. If not, the authors might need to dig into the published transcript data of pyrimethamine on the protozoan. By comparison, the molecular mechanism of lumefantrine would be cleared.
Response: Thank you for your good suggestions. (1) In our previous study [1], we have used sulfadiazine as a control drug to verify the efficacy of benzol in the treatment of Toxoplasma, and the results demonstrated that lumefantrine at 1.563μg/L yielded an inhibitory effect to intracellular proliferation of T. gondii comparable to 10 mg/L of sulphadiazine. In addition, our result also showed that lumefantrine curtailed the growth of T. gondii RH growing in Vero cells in vitro and prolonged the survival time of infected mice, so lumefantrine may be a promising agent to treat toxoplasmosis. However, despite its significance, the mode of action of lumefantrine is still unknown. Therefore, in this study, our aim is to explore the mechanisms of lumefantrine inhibition of T. gondii. So, based on our previous research foundation and results, in this study, we directly explored the mechanism of the inhibitory effect of lumefantrine on T. gondii proliferation by a metabolomic and transcriptomic study. (2) Both sulphonamides and pyrimethamine prevent the synthesis of folate by inhibiting the dihydrofolate reductase and dihydropteroate synthase that are essential for the survival and multiplication of parasites [2,3].Therefore, the mechanisms of action of the two drugs are well established.
Reference:
[1]WangDW, Xing ME, Saeed EA, Ding YY, Zhang X, Sang XY, Feng Y, Chen R, Wang XY, Jiang N, Chen QJ, Yang N. Determination of lumefantrine as an effective drug against Toxoplasma gondii infection – in vitro and in vivo study. Parasitology, 2021,148, 122–128.
[2]Derouin F. Anti-toxoplasmosis drugs. Current Opinion in Investigational Drugs, 2001, 2, 1368–1374.
[3]Anderson AC. Targeting DHFR in parasitic protozoa. Drug Discovery Today, 2005, 10, 121–128.
2.For RNA-seq, please specify how gondiiwas grown and harvest (Line 138, Line 401). How was the host RNA from Vero cell removed?
Response: Thank you for your good suggestion. We have added the steps of experimental operation on line 414-420 in the revised manuscript. Briefly, we scrapped off the cells from the flask bottom, syringed the cell and media mixture to break cells to release parasites. At this stage, the parasites are released but the host cell pieces are still much bigger than the parasites. Then, we passed the mixture through 5 μm of filter to obtain the parasites.
3.Please add scale bars to Figure 1E, even though the microscopic magnification was mentioned.
Response: Thank you for your good suggestion. We have added the scale bars to Figure 1E in the revised manuscript.
4.How the qPCR data were normalized? An internal housekeeping gene was required in each samples to normalize the levels of your genes of interest.
Response: Thank you for your good suggestion. In this study, the internal housekeeping gene is GAPDH [1] , please see “4.4. In vitro Proliferation Assay ” in the manuscript.
Reference [1] :Wang DW, Xing ME, Saeed EA, Ding YY, Zhang X, Sang XY, Feng Y, Chen R, Wang XY, Jiang N, Chen QJ, Yang N. Determination of lumefantrine as an effective drug against Toxoplasma gondii infection – in vitro and in vivo study. Parasitology, 2021,148, 122–128.
5.The PCA plot in Figure 2A only had three samples for each group. To increase the power, it requires at least 4 samples in each group. Someone might argue that the left four points are grouped tighter more closer than the two sample on the right. It is also more informative if the authors included at least one more treatment over the time scale. This would help us know which genes are first impacted by the drug treatment.
Response: Thank you for your good suggestion. We appreciate your thoughtful and constructive feedback. We agree that including one more sample could provide a stronger argument. From statistical standpoint, three samples are enough to make an argument especially when the three samples were clearly separated from the controls and the within group variation was small as shown in our graph.
6.Please make the font size bigger for Fig. 2CDE, Fig. 3A, Fig. 4 CDE. Especially the heatmap scale bar in Fig 4E.
Response: Thank you for your good suggestion. We have revised them in the revised manuscript.
7.It seems that lumefantrine affected the DNA replication system of the protozoan but why did it not affect Vero cells (Fig 1)? Can the authors discuss a little bit more? Maybe the drug cannot get into the cell nucleus?
Response: Thank you for your good suggestion. We appreciate your thoughtful question. At the concentration (900 ng/mL) that lumefantrine was effective at killing the parasites, we didn’t detect significant cytotoxicity. We agree with you that higher concentration of lumefantrine very likely will cause significant cytotoxicity. The molecular mechanisms of DNA damage we observed in our study could be investigated in the future. Based on studies published in other protozoa, it’s possible that lumefantrine could induce nucleic acid oxidative damage, inactivate/inhibit proteins that participate in DNA replication and repair, and eventually induce apoptosis. We have added some discussions on line 333-343 in the revised manuscript.
8.Could the author also explained why the galactose metabolism was affected (increased or decreased) by lumefantrine? The DMEM medium seems not to contain galactose.
Response: Thank you for your good suggestion and thoughtful question. We detected the differentially accumulated metabolites of D-fructose and sucrose which were mapped to the galactose pathway. Cell biosynthesis and metabolism of T. gondii were altered after the lumefantrine treatment.
